# Depth from 2D Images: Development and Metrological Evaluation of System Uncertainty Applied to Agricultural Scenarios

**DOI:** 10.3390/s25123790

**Published:** 2025-06-17

**Authors:** Bernardo Lanza, Cristina Nuzzi, Simone Pasinetti

**Affiliations:** Department of Mechanical and Industrial Engineering, University of Brescia, Via Branze 38, 25123 Brescia, Italy; bernardo.lanza@unibs.it (B.L.); simone.pasinetti@unibs.it (S.P.)

**Keywords:** optical flow, uncertainty evaluation, measurement science, precision agriculture, computer vision

## Abstract

This article describes the development, experimental validation, and uncertainty analysis of a simple-to-use model for monocular depth estimation based on optical flow. The idea is deeply rooted in the agricultural scenario, for which vehicles that move around the field are equipped with low-cost cameras. In the experiment, the camera was mounted on a robot moving linearly at five different constant speeds looking at the target measurands (ArUco markers) positioned at different depths. The acquired data was processed and filtered with a moving average window-based filter to reduce noise in the estimated apparent depths of the ArUco markers and in the estimated optical flow image speeds. Two methods are proposed for model validation: a generalized approach and a complete approach that separates the input data according to their image speed to account for the exponential nature of the proposed model. The practical result obtained by the two analyses is that, to reduce the impact of uncertainty on depth estimates, it is best to have image speeds higher than 500–800 px/s. This is obtained by either moving the camera faster or by increasing the camera’s frame rate. The best-case scenario is achieved when the camera moves at 0.50–0.75 m/s and the frame rate is set to 60 fps (effectively reduced to 20 fps after filtering). As a further contribution, two practical examples are provided to offer guidance for untrained personnel in selecting the camera’s speed and camera characteristics. The developed code is made publicly available on GitHub.

## 1. Introduction

Outdoor depth estimation can be achieved in several ways, depending on the hardware and software adopted. For applications requiring accuracy and dense depth reconstruction, LiDAR devices are the best choice [1]. However, the price of the hardware is high according to the depth accuracy provided, and oftentimes, such devices require the users to move slower than desired to ensure the correct acquisition of the point clouds (e.g., to deal with light interference in the best way possible). In addition, LiDAR systems require users to adopt dedicated hardware to keep up with the acquisition and storage needs of the device. Luckily, not every outdoor application requires the data density provided by LiDAR systems. In these cases, a viable and low-cost choice is the adoption of depth cameras leveraging stereo vision. However, the quality of the output (both the color and depth images) is often insufficient, especially when employed in applications where the camera moves fast, introducing defects due to the shutter and acquisition speed. In the case of high-speed movements, a good color camera paired with high-end optics is necessary to cope with the speed of the moving scenario to be acquired. Starting from good-quality images, the depth estimation problem could be addressed by computer vision (CV) techniques. One of the most popular methods is optical flow (OF), first developed in 1981 [2,3]. OF is defined as the apparent motion of objects in a sequence of images caused by the relative motion between the scene captured in them and the camera. Therefore, the problem encompasses several variables such as ambient illumination, object texture, and difficult geometrical shapes for which occlusions may happen, producing incorrect OF estimates [4,5,6]. In general, OF is obtained as a map of vectors indicating, for each pixel in the image, the corresponding apparent motion and its intensity. OF is used to estimate the speed and depth of both slow and fast phenomena and is applied in several fields, such as healthcare [7], robotics and moving vehicles in general [8], industry [9], and agriculture [10,11]. Modern approaches for solving OF issues exploit Deep Learning (DL) to improve the estimates, especially in the case of depth estimation [12,13]. Other recent advancements in the field of monocular depth estimation are presented in [14,15,16,17,18,19,20,21,22,23,24]. These works extensively adopt DL models to estimate depth from monocular images without employing OF and are considered state-of-the-art by the CV community. However, their depth output is not stable and it is not based on a measurement, not to mention the computational requirements needed to run those models on mobile and embedded devices. Some of those models do not produce outputs in metric coordinates either, making it difficult for untrained personnel to use those models in real in-field applications for which near real-time computation is required. Focusing on the agricultural sector, however, the general problem of OF estimation can be simplified since the measurement environment is more constrained. Farming vehicles, such as tractors and fruit-picking robots, move linearly along the rows of the field at a constant speed, which is not higher than 5 km/h in the case of large and heavy tractors and close to 2 m/s for agricultural robots. In addition, modern tractors are being designed to be autonomously driven, requiring accurate tractor–row distance estimation to adapt to the specific field they work on [25].

This article describes a simplified version of OF inspired by the Structure from Motion (SfM) CV problem, which allows the reconstruction of complex 3D structures from 2D multi-view images of the scene collected by a moving camera [26]. Traditional SfM techniques are computationally intensive and hardware-demanding, typically adopted for applications such as land reconstruction, architecture, and construction. Therefore, in this paper, a custom model that is easier to understand for end-users was designed. The developed model was validated by means of a practical laboratory experiment mimicking the agricultural scenario, thus showing how the model performs at different camera speeds and relative camera–object depths. In particular, the focus of this article stands on the models’ validation and uncertainty analysis, which ultimately provides ready-to-use information for the end-user (e.g., the farmer) to both choose the right camera for the target application and design the measurement setup and constraints. In fact, it is generally challenging to couple depth estimates with measurement uncertainty, typically treated as a confidence measure [27] that does not adhere to the “Guide to the Expression of Uncertainty” manual [28,29]. Moreover, monocular depth estimation research is mostly rooted in the CV community, which focuses more on the software and mathematical aspects of the problem and less on practical needs for the untrained end-user to obtain a reliable depth measure, such as which hardware to choose according to specific characteristics and what are the limitations of the measurement setup to keep in mind. The findings of this work could be beneficial for several research areas, including yield estimation [30,31] and the wood–fruits–leaf canopy volume estimation from 2D images [32]. Finally, the code used for this work is made publicly available on GitHub at [33].

## 2. Materials and Methods

### 2.1. Equipment and Experimental Setup

The experimental setup was designed considering the agricultural scenario, in which tractors and other vehicles move linearly at a constant speed. This is a constrained scenario that allows for some simplifications. The idea is that, by mounting a camera on the moving vehicle, it is possible to acquire a sequence of images for which the temporal and spatial relationships are known. This is possible because (i) the vehicle’s speed is known thanks to its encoder, and (ii) the acquired frames are coupled with acquisition timestamps. In addition, by mounting the camera orthogonal to the vehicle movement direction so that it looks at the canopy, its reference frame primarily moves along its Y axis. By focusing solely on the Y–Z plane with the motion vector confined within it, these assumptions allow for a simplified model implementation.

A scheme of the acquisition setup used for this work is shown in Figure 1, depicting a robot, a color camera, and a custom-made target. The moving vehicle is simulated by a robot (Universal Robots UR10e) mounted upside-down in the workspace described in [34]. On the robot’s end-effector a color camera (GoPro Hero 11 Black) was mounted, tasked with the acquisition of color images during movement. The robot was programmed to move horizontally at 5 constant speeds ($S_{1}=0.25$ m/s, $S_{2}=0.50$ m/s, $S_{3}=0.75$ m/s, $S_{4}=0.94$ m/s, and $S_{5}=0.97$ m/s) with a horizontal travel distance of 1.45 m. The speeds tested were selected according to the common operating speed of farming vehicles. The camera has a 1/1.9” CMOS sensor with a resolution of 5599×4927 px and F2.5 lens aperture. To increase the camera frame rate, during acquisition the camera was set to a resolution of 2704×1520 px with an aspect ratio of 4:3, allowing it to acquire frames at 60 frames per second (fps). The lens distortion coefficients and the internal camera matrix (containing the position of the image center $C_{X}$ and $C_{Y}$ [px] as well as the focal length dimension in both directions $f_{X}$ and $f_{Y}$ [px]) were computed by performing a calibration procedure using a chessboard pattern [35]. It is worth noting that the pixels of the chosen camera have a square shape; hence, $f_{X}=f_{Y}=f$.

The custom-made target measurand is made of an aluminum bar (metallic support) placed at a height of H=1.3 m from the ground, on which 5 smaller bars of different sizes ($B_{1}=10$ cm, $B_{2}=20$ cm, $B_{3}=40$ cm, $B_{4}=55$ cm, and $B_{5}=80$ cm) were fixed orthogonally. The target was crafted to simulate agricultural rows where plants grow at different depths than the guiding canopy. The metallic support was positioned at 4 different locations from the robot reference frame (RF), $D_{1}=115$ cm, $D_{2}=135$ cm, $D_{3}=155$ cm, and $D_{4}=165$ cm. These distances were chosen to simulate different working conditions (e.g., vehicles moving at different distances from the plants growing in the row, and different row distances).

On top of each bar, an ArUco marker [36,37,38] of size 45×45 cm was positioned to be orthogonal to the camera during acquisition ($M_{1}$, $M_{2}$, $M_{3}$, $M_{4}$, and $M_{5}$). ArUco markers are square matrices of black and white cells that easily represent a location and an orientation at the same time according to the deformation of the matrix pattern. Each ArUco marker is unique and procedurally generated by the related library, which includes algorithms and functions to retrieve the information of a specific marker from an image. In this work, the ArUco markers serve as the depth ground truth. The ArUco OpenCV library was employed to extract the markers’ depth [36,37,38]. Images taken from the acquisition setup (with the used ArUco marker shown) can be found in Figure 2.

### 2.2. Data Acquisition

A total of 20 tests were conducted (4 target positions *D* ×5 camera velocity *S*). Each camera acquisition is composed of a positive and a negative camera movement along the *Y* axis. Instead of saving each acquired frame independently, a more computationally efficient solution was defined. A single video in MP4 format was recorded, with each video including frames related to a specific combination of *D* and *S*, for both movements along the +Y axis and −Y axis. The customized algorithm detailed in Section 2.4 extracts individual frames from the video, splits movements along +Y axis and −Y axis, and computes the distance from the camera (depth *Z* m) of each ArUco marker. Subsequently, the displacement along the Y axis (ΔY px) of the marker’s center is calculated between two consecutive frames to apply the OF algorithm and estimate the depth.

The number of frames in a single trial varies depending on the robot’s (and thus the camera’s) velocity, as the robot follows a fixed path, with an average of 300 frames. Overall, a total of 60,000 data points were collected (5 robot speeds × 2 robot movement directions × 4 relative distances of the metallic support from the camera × 5 ArUco markers at different depths × 300 frames).

### 2.3. Model Definition

The model definition is based on the scheme of Figure 3 where the projection of a moving point of interest onto the camera plane and the object plane is represented.

Imagine a point *P* (with coordinates relative to the world’s reference frame, WRF) that moves in the object plane along the Y axis from position $P_{0}$ to position $P_{1}$ (displacement equal to ΔY). In the camera’s reference frame (CRF), the projection *p* of the point *P* on the camera plane moves from position $p_{0}$ to position $p_{1}$ (displacement equal to δY). This displacement is corrected for lens distortion through a camera calibration process (such as [35] or similar approaches). Considering the object plane and the image plane placed at a distance from WRF equal to *z* and *f*, respectively, there are two similar triangles ($P_{0}-P_{1}-WRF$) and ($p_{0}-p_{1}-WRF$). Based on their similarity properties, the equation is obtained as follows:(1)$$\frac{\Delta Y}{\delta y}=\frac{Z}{f},$$
where ΔY is the displacement of the point in the WRF [m], δy is the displacement of the point in the image plane [px], *Z* is the depth corresponding to the point [m], and *f* is the focal length of the camera [px].

By applying Equation (Equation 1), knowing the camera focal length *f*, it is possible to estimate the point’s *P* depth (*Z*) based on *p* pixel displacement in the image plane δy, given its relative displacement with respect to the WRF ΔY.

To accurately calculate δy with respect to a specific point or object within an image, several challenges arise in detection and tracking. The detection challenges include reliably identifying specific points or objects despite varying lighting conditions and similar-looking objects. Ensuring consistency in detecting the same points across multiple frames is crucial for accurate displacement calculation.

In tracking, the main issue is maintaining the continuity of the detected object across successive frames, which requires robust algorithms capable of handling rapid movements, changes in scale, and rotation. Additionally, tracking algorithms may experience drift over time, leading to deviations from the actual position due to cumulative errors.

Advanced techniques are typically employed to address these challenges. Feature extraction methods provide robust features invariant to changes in scale, rotation, and illumination, aiding in detection and tracking. Optical flow (OF) algorithms calculate the apparent motion of objects between consecutive frames, directly providing δy by analyzing pixel displacement. Marker-based tracking, using fiducial markers, offers a reliable method as these markers are easily detectable and their positions can be accurately determined.

Integrating these methods improves the accuracy and reliability of δy calculations. This displacement can be related to the original speed of the point *P* tracked according to the formula S=V·T, where *S* represents the space traversed by the tracked point, *V* refers to the movement velocity, and $T=t_{2}-t_{1}$ refers to the time between two consecutive frames. Two versions of this relationship can be defined accordingly:(2)$$\Delta Y=V_{camera}\left[m/s\right]\cdot T,$$(3)$$\delta Y=V_{image}\left[\mathrm{px}/s\right]\cdot T,$$
where $V_{camera}$ is the point *P* velocity in the object plane, $V_{image}$ is the point *P* velocity in the image frame, and *T* is the time between two consecutive frames. Therefore, the speed of the object in the WRF ($V_{camera}$) is related to the apparent speed of the object in the CRF ($V_{image}$).

By substituting these relations into Equation (Equation 1) and simplifying for *T* (purposely considered equal for the two terms), *Z* can be obtained directly from point *P* velocities (in the object plane and in the image plane):(4)$$Z=\frac{V_{camera}\cdot f_{Y}}{V_{image}}.$$

This equation will be referred to as the “analytical model” definition.

In this work, the aim is to simplify the relationship for *Z* estimation using only the OF output, treating $V_{camera}$ as a constant rather than an unknown variable. By assuming a constant $V_{camera}$ during movement, the idea is to obtain a general parameter *K* that allows end users to estimate *Z* with sufficient confidence for the target application using the data output of OF to obtain $V_{image}$. Following this, the “experimental model” definition is as follows:(5)$$Z=\frac{K}{V_{image}},$$
where $V_{image}$ [px/s] is obtained through the OF algorithm and parameter *K* [m·px/s] is unknown.

### 2.4. Depth Computation

The v=1…20 videos of the tests (see Section 2.2) were processed leveraging a variety of open-source functions available on the OpenCV library. The processing code was developed in Python and is publicly available at the link provided in reference [33]. Each video was analyzed independently, and the frames contained in it were not saved on disk; instead, to save space, a data extraction procedure was applied to analyze the contents of the frames and save the outputs in separate files in CSV format.

Each video sequence *v* contains a set of frames *i* acquired at time $t_{i}$. Each frame was processed individually following a two-block operational procedure illustrated in Figure 4 (first block in blue, and second block in pink). The first block involves calculating the pose of the markers thanks to a set of image processing analyses (yellow block inside the blue block of Figure 4), followed by the second block, which focuses on computing the optical flow for each marker. Examples of the frames registered by the camera during the experiments are shown in Figure 5.

The image processing block (yellow) structure is the following:Conversion to grayscale. The image of the current frame $I_{i}$ is converted from color to grayscale to facilitate the subsequent operations.Brightness and contrast correction. The image’s contrast α and brightness β are optimized to improve the image’s quality. These parameters are defined to ensure that the markers can be detected with sufficient accuracy by the ArUco library functions. The best values were experimentally found and are equal to α=2 and β=5 according to the general illumination of the working conditions of the experimental setup. To read further about how these two parameters are defined and used in OpenCV, please refer to the official documentation in [39].ArUco markers finding. The five markers *m* in the image $I_{i}$ are identified using a specific set of functions in the OpenCV ArUco library [36,37,38].Computation of apparent depth. Using the ArUco library, the depth $Z_{i,m}$ of each marker is calculated, providing an estimate of the marker’s *Z* coordinate relative to the camera’s reference system.Computation of markers’ centers. For each detected marker *m*, its center coordinates on the image plane, $C_{i,m}=\left(x_{i,m}^{\prime },y_{i,m}^{\prime }\right)$, are calculated. Coordinates $x^{\prime }$ and $y^{\prime }$ are different from the 3D coordinates (X,Y,Z), which involve marker and camera calibration since they are solely related to the marker’s detection on the image plane.

Outputs of the first operational block are the ground truth coordinates of the ArUco markers acquired, $\left(x_{i,m}^{\prime },y_{i,m}^{\prime },Z_{i,m}\right)$ for each acquired image $I_{i}$.

The second operational block (pink) was defined to compute pixel displacements between consecutive frames through OF. To do so, starting from the intermediate output table, the OF algorithm was applied on pairs of images $I_{i-1}$ and $I_{i}$ (thus beginning the counter from image i=2), using as tracked markers the centers of the markers $C_{i,m}$ detected before. This produces another value called $OF_{i,m}$ representing the pixel displacement δY between consecutive images in px/mm.

For each video v=1…20 the overall procedure produces a table $Tab_{v}$ of N−1 rows × 11 columns (time instant $t_{i}$ plus the 5 Z-coordinates of the markers’ centers $Z_{i,m}$ and the 5 OFs of each marker’s center $OF_{i,m}$). Each $Tab_{v}$, containing the OF data stream over time, is saved in a CSV file.

### 2.5. Signal Synchronization and Filtering

Considering the analytical model definition described by Equation (Equation 4), in the experiments $V_{camera}$ was equal to the robot’s speed. However, even if the robot is theoretically actuated at a constant speed, some portions of its movement include acceleration and deceleration (at the start, during changing direction, and at the end of the movement). Ideally, the robot’s movement during the experiment and the camera acquisition should be synchronized. However, the robot signal was obtained from its encoders, which produce denser data compared to the data obtained from the image-based analysis of Section 2.4. Thus, the two data streams are not synchronized and contain a different number of points. To address this issue and obtain synchronized data streams, the following procedure was applied. Please note that in the following notation, we will refer to the data stream containing the robot speed over time using $R_{v}$ and to the image speed over time (computed as the OF of the 5 ArUco markers’ centers’ positions in the image plane) using $Tab_{v}$, for a specific acquisition video v=1…20:The robot’s signal obtained from the encoders represents its position in its coordinate reference system. The robot’s speed over time $R_{v}$ is obtained by computing the first derivative of the signal.$R_{v}$ and $Tab_{v}$ are filtered to remove portions where the robot was not moving and, consequently, to separate between its positive (+Y) and negative (−Y) movements. To do so, the algorithm searched for OF and speed absolute values in $Tab_{v}$ and $R_{v}$, respectively, below the 5th percentile of the overall values. This threshold, determined iteratively to minimize data loss while eliminating noise during stationary phases, ensures that even minimal detected movements are treated as stationary. These rows correspond to the initial and final moments of acquisition when the robot was not moving; hence, they are removed from both $Tab_{v}$ and $R_{v}$ (red portions in the top image of Figure 6).To find the instant when the robot changes its moving direction (from +Y to −Y), the software scans $Tab_{v}$ and $R_{v}$ to select the rows where OF and robot speed values change from positive to negative. The turning point (yellow portions in the top image of Figure 6) is identified as the point where the corresponding signal goes under the 5th percentile of the values in $Tab_{v}$ and $R_{v}$, respectively (without the elements already filtered out from the previous step). Then, the negative stream is rectified, obtaining a signal always in the positive quadrant for both OF and robot speed values.The robot’s signal is sub-sampled to the same number of points as the corresponding OF signal of the experiment. Then, for each data point of the OF data stream $Tab_{v}$, the algorithm searches the temporally nearest neighbor of the corresponding robot’s data stream $R_{v}$. The iterative procedure outputs several values as many points in the data streams and then computes their average $T_{shift}$, representing the temporal shift between the two signals. The temporal correction $T_{shift}$ is then applied to $R_{v}$.The two data streams include acceleration and deceleration components (i.e., not a constant speed) that must be removed to obtain only data corresponding to movements at a constant speed. This step is applied on both $Tab_{v}$ and $R_{v}$ at the same time. The points to be removed are selected iteratively by removing parts at the beginning and at the end of the original curve and computing the linear regression with the obtained curve. The procedure removes the initial and final 1% of the whole curve first and iteratively increases the removal percentage up to 20% (with steps of 1%). For each curve obtained, a linear fit is computed coupled with the corresponding $R^{2}$. The result is a function of the distribution of $R^{2}$ values with respect to the percentage of removed data. The ideal constant velocity segment corresponds to the portion of the whole curve with the greatest $R^{2}$. It was experimentally found that the ideal value for sub-sampling is 16% for all acquisitions since $R^{2}$ does not change notably afterward. An example of this procedure is graphically shown in the bottom image of Figure 6, where the program iteratively selects portions of the data (depicted with colored bars to highlight the portion of data considered, in pairs) until the optimal portion is selected (in the figure, this is the black one).

At the end of this procedure, the data in each $Tab_{v}$ is merged according to the robot’s speed. The result is a total of 5 tables called $Data_{s}$, comprehending, for each ArUco marker, all the $P_{s}$ captured points obtained during the experiment for that specific robot’s speed (subscript *s* refers to the five robot speeds, ranging from 1 to 5 and equal to $S_{1}=0.25$ m/s, $S_{2}=0.50$ m/s, $S_{3}=0.75$ m/s, $S_{4}=0.95$ m/s, and $S_{5}=0.97$ m/s). Please note that the number of $P_{s}$ depends on the robot’s speed because the quantity of captured points varies according to it (more points for slower speeds).

### 2.6. Window-Based Filtering

The data in $Data_{s}$ resulting from the processing described in Section 2.5 is significantly noisy, especially at higher speeds. This is primarily due to the vibrations of the camera and sudden environmental disturbances (e.g., light changes, electrical noise, and optical aberrations) that are inherent in real-world conditions, especially in agricultural scenarios where light scattering effects appear on the plants’ canopy. These factors introduce noise and uncertainty into the acquired images, which in turn affect the OF output, a well-known issue in the literature [4,5]. Moreover, in those scenarios, it is quite common to track specific objects during acquisition, for example, fruits [40,41], by using DL models such as object detectors, including in the uncertainty also the contribution of incorrect prediction or bounding box positioning. To mitigate this issue, the data in $Data_{s}$ are filtered again using a window-based moving average technique. In addition, this approach effectively reduces the impact of incorrect DL predictions in the case of fruit tracking.

The filter was applied on each $Data_{s}$, and considering the data of each ArUco marker individually, the best results were obtained by setting the window size equal to 3, resulting in a sub-sampling of the data corresponding to an “effective” camera acquisition speed of 20 fps (in contrast with the original 60 fps).

In the subsequent analysis, results will be shown for data with and without the application of the window-based moving average filter, for a total of k=1…10 (5 robot speeds $S_{1}...S_{5}$, with and without filtering) experimental models represented by tables $Data_{k}$. Accordingly, the quantity of captured points contained in each $Data_{k}$ will now be called $P_{k}$ since their number is further reduced after filtering. An example of the windowing effect can be seen in Figure 7.

### 2.7. Model Validation and Uncertainty Estimation

Each data set $Data_{k}$ contains a set of $p=1\ldots P_{k}$ points represented by a certain $V_{image}$ obtained as the OF value of the ArUco marker *m* (m=1…5), $OF_{i,m}$, and a certain depth value $Z_{i,m}$, computed for each acquired image *i* in the data set $Data_{k}$ as described in Section 2.4. For simplicity, the points of a data set are generally represented by the pair ($V_{p,k}$, $Z_{p,k}$), a notation that takes into account all *m* marker points contained in a specific $Data_{k}$. Therefore, after conducting the pre-processing and filtering steps described in Section 2, the resulting k=1…10 data sets $Data_{k}$ are used to verify if the experimental model of Equation (Equation 5) is in agreement with the analytical one in Equation (Equation 4). To do so, it is first necessary to obtain an estimation of the parameter *K* for each data set, namely $K_{k}$. For this purpose, the data set points are simply used as the input for a curve-fitting method implemented in Python 3.10 by the function “curve_fit” of the SciPy package [42], producing a $K_{k}$ for each data set. Using the known information about the robot speed (one of the 5 tested speeds in our experiment $S_{1}\ldots S_{5}$, which can be considered equal to $V_{camera}$ since the camera is rigidly mounted on the robot’s end-effector), the focal length of the camera ($f_{Y}$), and the acquired $V_{image}$ of each point ($V_{p,k}$), it is straightforward to also apply the analytical model in Equation (Equation 4) and compare the two. Figure 7 shows the resulting comparison of the two models for the original data and for the filtered data corresponding to the robot speed $S_{3}=0.75$ m/s (the results for the other robot speeds are similar and are omitted for brevity). In both cases, the experimental and analytical models are overlapped (red dashed line versus black solid line), meaning that the experimental model was correctly defined and that the experimental procedure was properly conducted. However, by observing the curves, a major issue arises due to the exponential nature of the models. In fact, data points corresponding to $V_{image}$ close to 0 px/s are distributed over a wider range of possible values, resulting in high variability. This effect is strongly reduced when $V_{image}$ is higher than 500–800 px/s according to the camera’s speed, for which the model shows acceptable variability for the target application. Due to this issue, the problem of uncertainty estimation of the experimental model is challenging to tackle. In the following analysis, two approaches are proposed to obtain the model’s uncertainty: a “generalized approach” and a “complete approach”. A scheme of the analysis conducted to estimate the model’s uncertainty is shown in Figure 8.

Both approaches are based on a common starting point based on a Monte Carlo generation of simulated data points. The procedure, called “Monte Carlo generation” in Figure 8, is as follows:Original data distributions definition. In real-world applications, the values of each point’s speed and depth are affected by uncertainty. To validate the proposed experimental model, their variability was empirically set to $\sigma_{V}=1$ px/s and $\sigma_{Z}=0.005$ m, respectively. These two values were estimated considering the overall data acquired and the ArUco markers documentation [36,37,38]. Now, considering a certain set of points $Data_{k}$, for each point ($V_{p,k}$, $Z_{p,k}$) contained in it, two Gaussian distributions were built using the data point’s actual values as the mean, $\mu_{V}$ and $\mu_{Z}$, and the variabilities defined before, $\sigma_{V}$ and $\sigma_{Z}$, as the distribution’s spread.Synthetic data generation. From the distributions of *V* and *Z*, a total of 10,000 simulated data points (${\hat{V}}_{p,k}$, ${\hat{Z}}_{p,k}$) were generated for each original point. This process produces a table $Synth_{k}$ composed of 10,000 rows and $P_{k}$ columns. The synthetic data generation procedure was repeated for each tested robot’s velocity $S_{1}\ldots S_{5}$, for both original and filtered data, obtaining a total of k=1…10 tables $Synth_{k}$.Estimation of parameter K. Each row r=1…10,000 of $Synth_{k}$ now contains $P_{k}$ synthetic points. Therefore, it is possible to estimate parameter *K* for that specific row, $K_{r,k}$, by fitting the experimental model to the data in the row. Repeating this process for all rows produces 10,000 estimated values of $K_{r,k}$.

### 2.8. Uncertainty Estimation Using the Generalized Approach

The following steps are conducted after the “Monte Carlo generation” procedure described in Section 2.7 and are graphically shown in Figure 8 in the block called “Generalized approach”.

Given the 10 tables $Synth_{k}$ and the corresponding $K_{r,k}$ for each row r=1…10,000, for each point $p=1\ldots P_{k}$ in the row, it is possible to compute ${\tilde{Z}}_{r,p,k}$, which is the depth value output produced by applying the experimental model in Equation (Equation 5). Then, residuals $\varepsilon_{r,p,k}$ of each point are calculated as the absolute difference between the synthetic depth ${\hat{Z}}_{r,p,k}$ and the depth obtained from the experimental model, ${\tilde{Z}}_{r,p,k}$:(6)$${\tilde{Z}}_{r,p,k}=\frac{K_{r,k}}{{\hat{V}}_{r,p,k}},$$(7)$$\varepsilon_{r,p,k}={\hat{Z}}_{r,p,k}-{\tilde{Z}}_{r,p,k},$$
where ${\hat{Z}}_{r,p,k}$ and ${\hat{V}}_{r,p,k}$ are the depth and the image speed of a synthetic data point *p* in row *r*, respectively, and $K_{r,k}$ is the experimental model parameter *K* estimated for each row r=1…10,000 of the data set $Data_{k}$ at the end of the “Monte Carlo generation” procedure.

Then, for each row *r*, it is possible to compute the sample standard deviation $\sigma_{r,k}$ as(8)$$\sigma_{r,k}=\sqrt{\frac{\sum_{p}^{P}\varepsilon_{r,p,k}^{2}}{P-1}},$$
where *P* corresponds to the original number of data points $P_{k}$ in the columns of table $Synth_{k}$.

Finally, we compute the root mean square (RMS) of all the $\sigma_{r,k}$ values, which is the final uncertainty for each model, $u_{k}=RMS_{k}$:(9)$$RMS_{k}=\sqrt{\mathrm{mean}(\sigma_{r,k}^{2})}.$$

### 2.9. Uncertainty Estimation Using the Complete Approach

The following steps are conducted after the “Monte Carlo generation” procedure described at the start of Section 2.7 and are graphically shown in Figure 8 in the block called “Complete approach”.

After generating the 10 tables $Synth_{k}$ and obtaining the r=1…10,000 model parameters $K_{r,k}$, the experimental model in Equation (Equation 5) is now estimated for each robot speed, for a total of k=1…10 models, $Model_{k}$. Using as the input the estimated $K_{r,k}$ and the original image velocity data of each point $V_{p,k}$ (given from the OF), the model outputs a depth value ${\tilde{Z}}_{p,k}$. Repeating this step for all the points in all $Data_{k}$ tables produces new data tables $Model_{k}$, in which each point is described as the pair ($V_{p,k}$, ${\tilde{Z}}_{p,k}$).

As already discussed at the start of Section 2.7, given the exponential nature of the model, the data points for which *V* was close to 0 px/s demonstrated a behavior very different from those belonging to the latter part of the graph where *V* was close to 3000 px/s. To analyze this issue, the data points have been split into groups according to their $V_{p,k}$ value. For all $Model_{k}$ tables, the data points were divided into groups according to the value of $V_{p,k}$. By considering the range of possible values of $V_{p,k}$ for that specific $Model_{k}$, groups were created with a step of 30 px/s (e.g., group 1 contained points with $V_{p,k}$ in the range [10, 40) px/s, group 2 with $V_{p,k}$ in the range [40, 70), etc.). This resulted in a certain number of groups g=1…G according to the specific table $Model_{k}$ considered. The $RMS_{k}$ values described by Equation (Equation 9) are now calculated on the data belonging to each group *g* for each model *k*; namely, $\sigma_{g,r,k}$ is obtained by using Equation (Equation 8) and $\varepsilon_{g,r,p,k}$, as in Equation (Equation 6). This produces group-separated RMS values for the specific experimental model *k* considered, $RMS_{g,k}$. The final uncertainty in this approach is composed of two contributions for each model *k* and it is separated by velocity group *g*: one is given by the $RMS_{g,k}$, and the other is given by the standard deviation $\sigma_{{\tilde{Z}}_{g,k}}$ of the depth values obtained by applying the experimental model and is divided by group, namely ${\tilde{Z}}_{g,p,k}$. The computation of the group-separated uncertainty for each model *k* is given by the following:(10)$$u_{g,k}=\sqrt{RMS_{g,k}^{2}+\sigma_{{\tilde{Z}}_{g,k}}^{2}}.$$

## 3. Results and Discussion

The resulting uncertainties for the generalized approach are shown in Table 1 for both original and filtered data. The effect of the window-based filter is evident for $S_{1}$ and $S_{2}$, for which the overall uncertainty is reduced by 50%, while for $S_{3}$ the effective uncertainty reduction is only 20%. In the case of $S_{4}$ and $S_{5}$, the effect is even less evident, reducing the uncertainty of just 0.01 m in both cases (5%). This effect is related to the number of frames acquired according to the robot’s speed; in fact, since the camera’s acquisition rate is the same in all tests (60 fps), at lower speeds, more images are acquired representing the same scene; thus, there is not sufficient displacement in between two consecutive pictures for OF to work well. By sub-sampling the data, the effect is to virtually reduce the camera’s acquisition rate to 20 fps, incrementing the spatial difference between two consecutive frames and thus improving OF estimation. In addition, after applying the filtering, the overall uncertainty for $S_{1}=0.25$ m/s is the same as the one obtained for $S_{3}=0.75$ m/s. Generally, best-case scenarios are obtained for $S_{2}=0.50$ m/s and for $S_{3}=0.75$ m/s. These speeds are acceptable for the agricultural scenario considered. Figure 9 shows the histograms of the distribution of values $\sigma_{r,k}$ from which the final uncertainty $RMS_{k}=u_{k}$ is computed, for the best-case scenarios (models with $S_{2}$ and $S_{3}$), both with no filter and with a filter applied. All the other models were omitted for brevity since the resulting histograms are the same.

As for the complete approach, the results of $RMS_{g,k}$ and $\sigma_{{\tilde{Z}}_{g,k}}$ are shown in Figure 10a and Figure 10c, respectively, for models with k=1…5 (no filtering), and in Figure 10b and Figure 10d, respectively, for models with k=6…10 (filtering applied). Obviously, the groups of each model are not comparable since they depend on the numerosity of points belonging to the group, which in turn depends on the total number of points $P_{k}$ in the original data set $Data_{k}$. In Figure 10e,f the contribution of $RMS_{g,3}$ and $\sigma_{{\tilde{Z}}_{g,3}}$ towards the computation of the total uncertainty $u_{g,3}$, for $S_{3}=0.75$ m/s in both the unfiltered and filtered cases (models with k=3 and k=8, respectively), is displayed. The results for the other robot speeds $S_{1}$, $S_{2}$, $S_{4}$, and $S_{5}$ are similar, so they were omitted for brevity. It is evident that almost all the contribution is due to $RMS_{g,k}$, and its trend indicates that estimating the depth of a given point from OF is not robust in the first area of the graph where the speed of the point (OF) is lower than 500–800 px/s according to the camera’s speed. However, for point speed higher than this value, the overall uncertainty is reduced to less than 20 cm.

Measurement uncertainty on the computation of depth is even less than 100 mm for robot speeds equal to $S_{2}=0.50$ m/s and $S_{3}=0.75$ m/s, while it increases for the other three (see Figure 10a,c). This is interesting because it highlights that moving at a very low speed ($S_{1}=0.25$ m/s =0.9 km/h) gives similar uncertainty values to those obtained when moving at higher speeds ($S_{4}=0.95$ m/s =3.4 km/h and $S_{5}=0.97$ m/s =3.5 km/h). This effect can be explained by how depth is estimated from OF. When the robot moves too slowly ($V_{camera}$ is too low), there is not sufficient pixel difference between consecutive pairs of images for OF to produce an accurate estimation of $V_{image}=OF_{i,m}$, while the accuracy of the ArUco markers’ apparent depth $Z_{i,m}$ is higher with lower uncertainty due to a more stable image frame. On the other hand, when the robot moves faster, the estimation of $V_{image}$ improves until the amount of pixel difference between consecutive images is too high for OF to produce a valid estimation since the two images could be so different from each other that the point matching fails. Consequently, the accuracy in the estimation of apparent depth becomes less accurate when $V_{camera}$ increases. Evidently, by applying the window-based filtering, the variability of OF estimation is reduced, and the overall depth estimation is improved despite losing data points (see Figure 10b,d). This drawback is acceptable for the target application (agriculture) and other applications for which sampling one data point in every three is not an issue.

To conclude, the sub-sampling of data points is closely related to the camera acquisition speed (fps): given the issue of OF not being able to produce reliable outputs when the image pairs’ difference is too low, adopting cameras with high fps is not the best choice. On the other hand, considering the possibility of sub-sampling the acquired frames, choosing a camera with less than 30 fps could lead to similar issues. In both cases, the outcome may be similar to the model produced for $S_{1}$. It is worth noting that a few points are missing in the region of 2000–2300 px/s in Figure 10e,f. This is due to poor detection of some markers happening for $V_{3}$ and the lack of detected points for that $V_{image}$ speed range in the case of $V_{1}$ and $V_{2}$, as can be noted in Figure 10a,b. Poor detections may occur if the acquired image is too blurred (as is the case for $V_{3}$) or if illumination conditions create aberrations on the ArUco markers, resulting in incorrect or missing detections. However, considering the points present for the speed range higher than 2300 px/s, it is safe to say that the overall trend of the two curves in Figure 10e,f is unaltered.

### 3.1. Limitations of the Method

It is worth stressing that the present article describes the theory of the measurement principle validated through a laboratory experiment. Therefore, the choices made to design the setup and its constraints (Section 2.1) refer to agricultural scenarios in general but can be easily generalized to other environments as well (e.g., industry and autonomous driving). An important limitation of this approach is that the validation was only conducted indoors; hence, other impacting parameters typical of the outdoor environment were not studied in detail. These are as follows: (1) The illumination of the scene, which must be sufficient to see the points of interest in the image but, at the same time, not too bright to avoid aberrations or light spots that may interfere with the recognition of the targets. This is especially important since bright spots and blurred areas of the targets produce incorrect or missing detections. To this aim, an interesting design choice is the addition of an illumination system, for example, in the case of industrial or dark scenarios, to improve the clarity and contrast of image details. (2) The eventual vibrations of the moving vehicle, which may produce incorrect estimations of the OF values and should therefore be compensated either during acquisition or afterwards. (3) Adverse meteorological conditions that may interfere with data acquisition, including dirt. However, assessing the impact of these factors on the method and eventually proposing algorithmic solutions to overcome them requires a more in-depth outdoor study, which will be conducted in the future as a further advancement of the proposed method.

Concerning the inference time, e.g., the time required for a new pair of consecutive frames to be processed by the proposed model and pipeline to produce a final depth estimation, some considerations should be made. (1) The inference can be applied only to pairs of frames because OF is calculated on their differences; thus, we should consider the time required by the camera to acquire and save two frames (either on disk or in volatile memory). This may be a crucial bottleneck, especially if the acquisition hardware is an embedded platform, slowing down acquisition to even 10 fps instead of the typical 30 fps of most common devices. (2) At least one point of interest should be detected in both images of the pair. In the case of our experiment, the points of interest were the ArUco markers, but for other applications, it can be something else. This detection requires dedicated software, which may be AI-based instead of classical computer vision techniques (as is the case for the ArUco markers library adopted in this work). The ArUco markers were detected at a speed of approximately 100 ms per image. Keep in mind that AI-based software is often slower, requiring even 500 ms per image. (3) OF should be computed on the points of interest detected in the pair of images, which is a fast and well-known algorithm that benefits from GPU acceleration if available, requiring only a few ms per pair of images. (4) The final step is the application of the proposed model to obtain the depth estimation in meters according to the estimated $V_{image}$ speed obtained, which is a simple formula requiring a few ns to be computed. All things considered, we experimentally found that for our setup the average inference time is approximately 300 ms.

### 3.2. Practical Examples

In the next sections, two examples provide some practical conclusions for agronomists and researchers aiming at conducting on-the-go depth measurements using the proposed method, taking into consideration the uncertainty behavior explained above. In particular, the examples explain (1) how to choose the correct vehicle speed given a specific camera, and (2) how to choose the correct camera for the target application given the vehicle speed ($V_{image}$).

#### 3.2.1. Choose the Correct Vehicle Speed Given a Specific Camera

Let us consider a camera with a sensor size of $S_{X}\times S_{Y}=7.2\times 5.4$ mm, pixel resolution of $PR_{X}\times PR_{Y}=1600\times 1200$ px, and 60 fps acquisition speed. The camera is fixed on a rigid case mounted on the side of a vehicle, so the pixels vary along the *X* axis of the camera (assuming negligible vibrations, so no movement along *Y*). From the results obtained and discussed in the previous section, it is assumed the minimum and maximum acceptable image velocity are $V_{image}^{min}=500$ px/s and $V_{image}^{max}$ to keep measurement uncertainty on estimated depth values acceptable for the target application.

Consider adopting a maximum vehicle speed equal to $V_{vehicle}^{max}=4\cdot V_{vehicle}^{min}$. In these conditions, the camera obtains a pair of frames after a time dT=1/fps=1/60 s, so the amount of pixels that change in two consecutive images according to the chosen speed is calculated as $C_{px}=V_{image}\cdot dT$. Using the minimum and maximum image speeds, it results that $C_{px}^{min}\approx 8$ px and $C_{px}^{max}\approx 33$ px.

Considering the target application of an agriculture scenario where a tractor moves in between two rows, it is known that the inter-row distance is typically set to 2 m, so it can be reasonably assumed that the working distance WD, which is the camera–object distance, is WD≈800 mm. By using the following formula, we can calculate the camera’s field of view (FoV) along the *X* axis:(11)$$FoV_{mm}=\frac{S_{X}\cdot WD}{f},$$
where *f* is the focal length of the chosen optics. Several options exist on the market, and the choice depends on the magnification effect desired according to the operating WD, recalling from the basics of digital photography [43] that higher values of *f* correspond to a higher magnification and a narrow FoV, while lower values correspond to the opposite case. In this example, let us assume f=8 mm, corresponding to $FoV_{mm}=720$ mm. Now it is possible to calculate the millimeters-to-pixels resolution ratio $R_{s}$ as follows:(12)$$R_{s}=\frac{FoV_{mm}}{PR_{X}}.$$

A low value of $R_{s}$ corresponds to a higher number of pixels needed to represent a millimeter, thus producing image details with a higher number of pixels. Having a sufficient number of pixels to represent the smallest target object in the image is key to successful computer vision applications. For this example, it results in $R_{s}=0.45$ mm/px. Then, the minimum and maximum vehicle speeds can be obtained using the following formulas:(13)$$V_{vehicle}^{min}=R_{s}\cdot V_{image}^{min},$$(14)$$V_{vehicle}^{max}=R_{s}\cdot 4\cdot V_{image}^{min}.$$

This results in $V_{vehicle}^{min}=225$ mm/s =0.23 m/s =0.81 km/h and $V_{vehicle}^{max}=0.90$ m/s =3.24 km/h. Given the described camera, the tractor must maintain its velocity between $V_{vehicle}^{min}$ and $V_{vehicle}^{max}$ to obtain the most reliable depth measurements.

#### 3.2.2. Choose the Correct Camera for the Target Application Given the Vehicle Speed

In the second scenario, the vehicle speed and the working distance are already defined and the question is about the choice of the right camera for the application. For the sake of the example, let us consider $V_{vehicle}=5$ km/h =1.39 m/s =1388.89 mm/s and WD=1000 mm. Hence, by inverting Equation (Equation 13) and considering $V_{image}^{min}=500$ px/s (which is the optimal image velocity resulting from our experiments), the result is $R_{s}=2.78$ mm/px. Let us assume that, for the target application, at least $FoV_{mm}=1000$ mm must be viewed by the camera. Inverting Equation (Equation 12) results in $PR_{X}=360$ px. The end-user is now encouraged to search among the plethora of available cameras on the market with a pixel resolution along *X* at least equal to 360 px, a requirement easily fulfilled nowadays. After selecting the best product for its needs, it is straightforward to also select the optics by using Equation (Equation 11) to find the value of *f*. Once again, the resulting value is the optimal one obtained from mathematical formulations; thus, it may differ from what is available on the market. The process of finding the right camera and optics may be a long one since it requires adjustments and comparisons with several products.

Finally, two considerations should be made about possible aberrations appearing in the acquired frames when using cameras for on-the-go acquisitions. The first issue is motion blur, an effect appearing when the vehicle moves too fast and the camera acquires too slowly. In this case, it is recommended to use cameras with high fps. The second issue is about the camera’s shutter. Low-cost cameras are typically equipped with a rolling shutter; however, this results in the upper and bottom portions of the image corresponding to two different scenes, an issue especially for fast movements (a typical example is the picture of a fan). To avoid this issue, it is recommended to choose cameras equipped with a global shutter.

## 4. Conclusions

The presented work deals with the topic of depth estimation from a moving monocular 2D camera, leveraging optical flow. Starting from the classic analytical model used to estimate depth from moving images, an experimental model that is easier to apply for end-users is proposed and validated. The experimental setup comprises a robot that simulates the moving vehicle, on which the camera is mounted. The target measurand is a rigid frame with five bars of different lengths mounted on it to simulate objects positioned at different depths. On top of each bar, an ArUco marker was fixed. A total of five experiments were conducted by actuating the robot at five speeds, each time recording a video that contains both positive and negative robot motions. The developed software analyzes the videos to extract the ArUco markers’ apparent depth (*Z*) and compute the optical flow ($V_{image}$) from pairs of images. A window-based moving average filter was developed and applied to the acquired data to reduce noise and improve the final uncertainty.

The core of this work lies in the metrological validation of the system and the computation of measurement uncertainty, for which two approaches are proposed as follows: the generalized approach and the complete approach. In the case of the generalized approach, the best-case scenario is obtained for robot speeds equal to $S_{2}=0.50$ m/s and $S_{3}=0.75$ m/s, for which the corresponding uncertainty on depth estimation is $u_{2}=0.08$ m (no filter) and $u_{7}=0.04$ m (filter applied) for $S_{2}$, and $u_{3}=0.09$ m (no filter) and $u_{8}=0.07$ m (filter applied) for $S_{3}$. The complete approach separates the depth data according to their speed $V_{image}$ to deal with the exponential nature of the models, producing group-separated results for each model with and without filtering. Again, the best case scenario is obtained for robot speed equal to $S_{2}$ or $S_{3}$.

Another interesting conclusion useful for end-users is that the experiments highlighted that low image speeds increase the total uncertainty when the pixel displacement between two consecutive images is insufficient, thus reducing the robustness of the optical flow algorithm. This effect directly impacts both the vehicle speed and the camera’s acquisition rate. Generally, it is shown that for image speeds higher than 500–800 px/s, uncertainty on depth estimation drops below 200 mm. This outcome is especially useful for applications such as on-the-go depth measurements in scenarios where 100 to 200 mm uncertainty is acceptable, such as agriculture. In fact, in this specific context, it is common to have low-cost cameras mounted on moving vehicles and the main questions are as follows: “Given a certain camera, at which speed should the vehicle move to get stable depth readings?” and “Which camera should be bought if the vehicle moves at a certain speed?” To answer these questions, two practical examples are presented and discussed, showing how the proposed work can be effectively employed by end-users in the two most common scenarios.

To conclude, the presented work is a stepping stone towards the development of reliable, easy-to-use, and low-cost embedded measuring systems suitable for in-field measurements for a plethora of applications, especially in agriculture. Future work will be devoted to the optimization of the presented methodology by combining it with modern Deep Learning models; for example, for fruit counting and measurements.

## Figures and Tables

**Figure 1 sensors-25-03790-f001:**
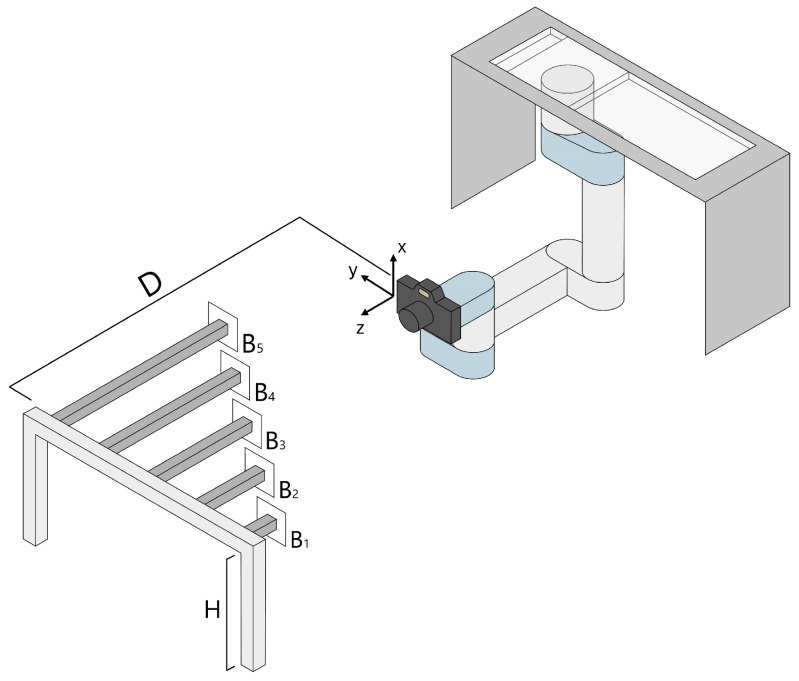
Graphical scheme of the setup adopted in this work, highlighting relevant parameters and sizes of the involved equipment.

**Figure 2 sensors-25-03790-f002:**
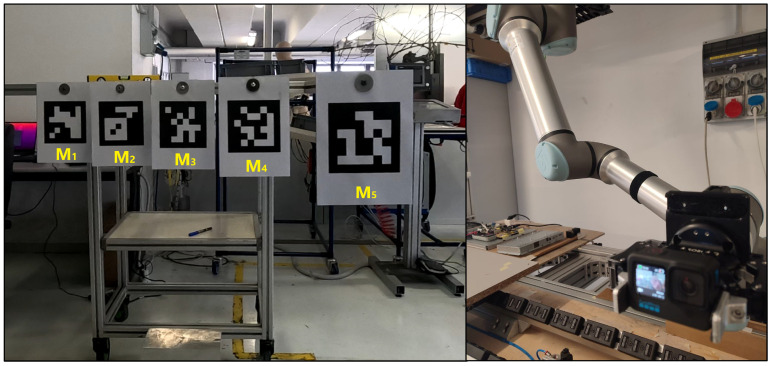
Example of the setup and the equipment adopted in this work. ArUco markers are highlighted in yellow.

**Figure 3 sensors-25-03790-f003:**
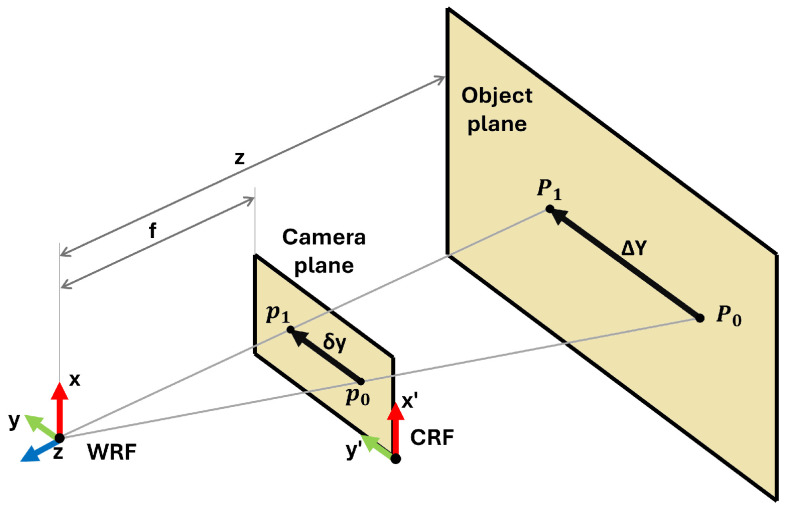
Scheme depicting the projection of a point from the real world to the image plane assuming a pinhole camera model.

**Figure 4 sensors-25-03790-f004:**
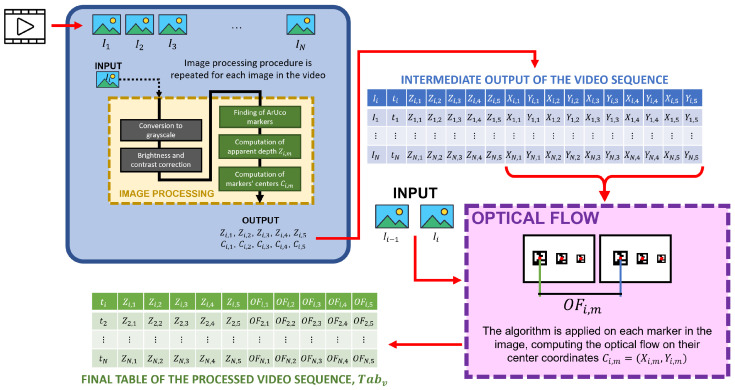
Graphical scheme of the processing procedure divided into two operational blocks (blue and pink). The image processing step (yellow) is repeated for each image i=1…N in the video sequence *v*, producing an intermediate output table. Then, the OF algorithm is applied to image couples, starting from image i=2 and tracking changes in the markers’ centers (obtained from the first operational block in blue).

**Figure 5 sensors-25-03790-f005:**
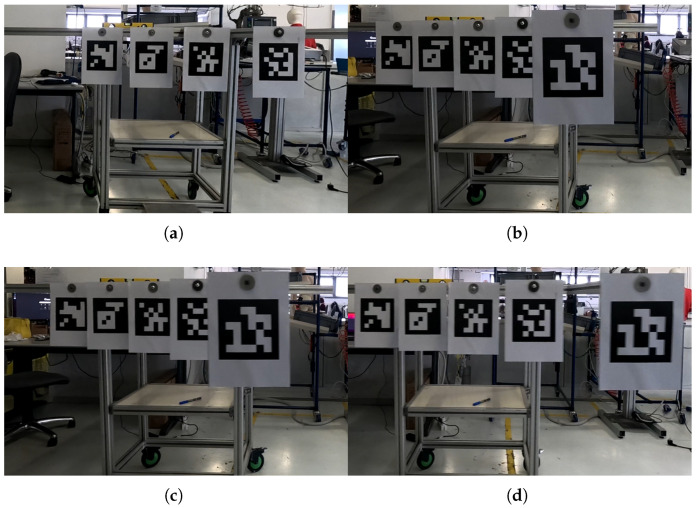
Examples taken from the video sequences *v* at different speeds. Please note that the closer marker ($M_{5}$) moves outside the camera’s field of view when the robot is in the leftmost end position, due to the setup’s positioning in the laboratory with respect to the available space. (**a**) $S_{1}=0.25$ m/s, (**b**) $S_{2}=0.50$ m/s, (**c**) $S_{3}=0.75$ m/s, (**d**) $S_{5}=0.95$ m/s.

**Figure 6 sensors-25-03790-f006:**
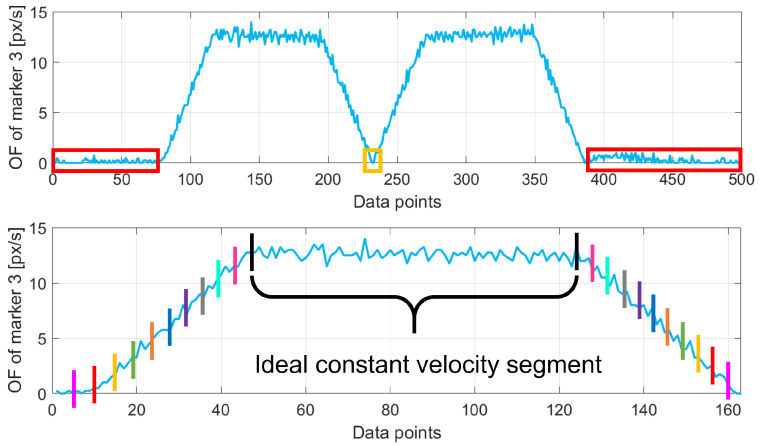
Graphical example of the filtering procedure applied on the data of the ArUco marker $M_{3}$ at speed $S_{1}=0.25$ m/s. (**top**) Removal of the portions with no robot movement (red) at the initial and final stages of the movement and corresponding to the change in direction (yellow). (**bottom**) Example depicting the selection process of the ideal portion of data corresponding to a movement at constant velocity.

**Figure 7 sensors-25-03790-f007:**
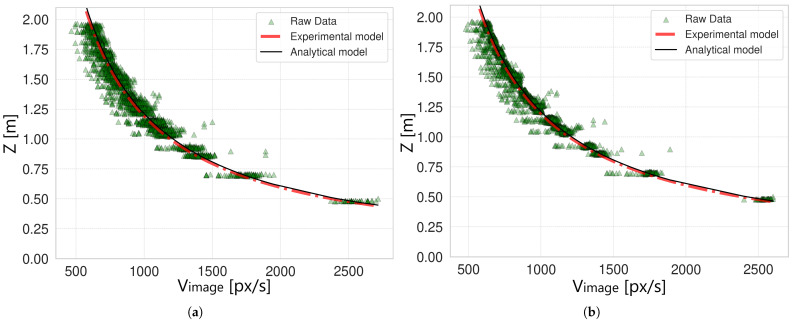
Comparison between the analytical model (black line) and the experimental model (red line) for robot speed $S_{3}=0.75$ m/s. The experimental model is fitted to the points to obtain parameter *K*. The green triangles correspond to (**a**) the acquired marker points, and (**b**) the filtered points.

**Figure 8 sensors-25-03790-f008:**
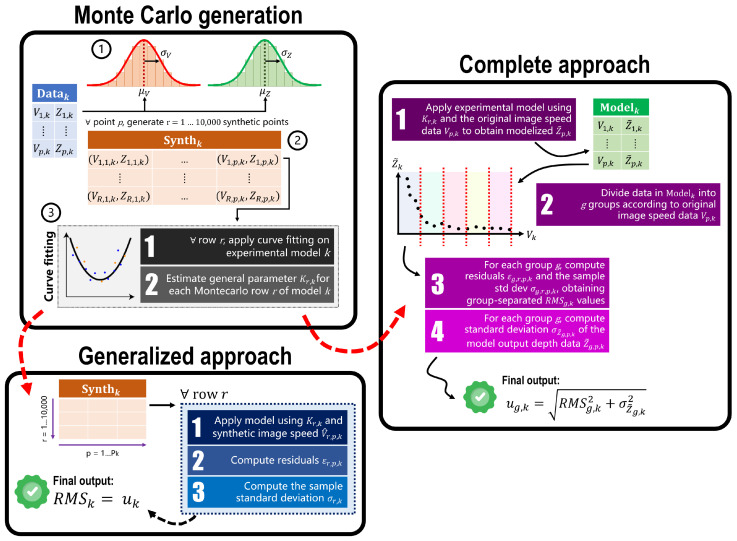
Graphical scheme of the uncertainty estimation procedure, divided into three blocks: (1) Monte Carlo generation, which is the starting point for both consequent approaches, (2) the generalized approach, giving a general uncertainty value for each tested model *k*, and (3) the complete approach, which computes uncertainty values for each tested model *k* by also dividing data points by velocity group, thus accounting for their different characteristics.

**Figure 9 sensors-25-03790-f009:**
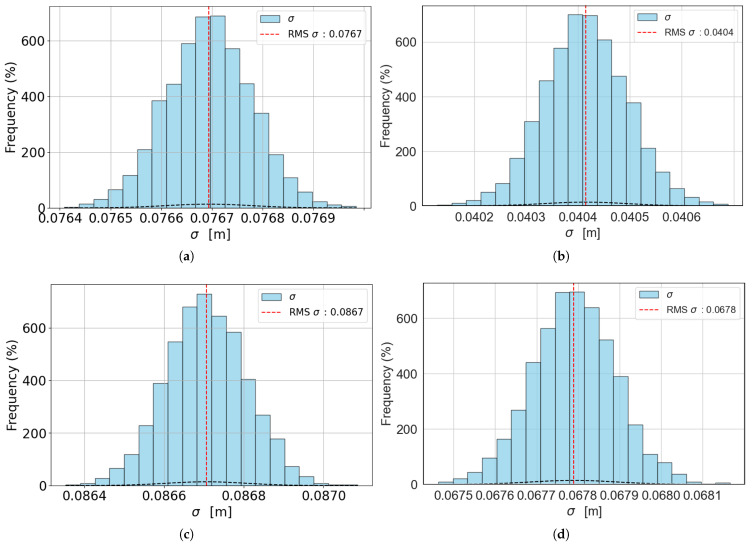
Histograms showing the distribution of values $\sigma_{r,k}$ (blue bars) and the resulting $RMS_{k}$ (red dashed line). (**a**,**b**) Histograms of models with k=2 and k=7 ($S_{2}=0.50$ m/s) with no filtering and filtering applied, respectively. (**c**,**d**) Histograms of models with k=3 and k=8 ($S_{3}=0.75$ m/s) with no filtering and filtering applied, respectively.

**Figure 10 sensors-25-03790-f010:**
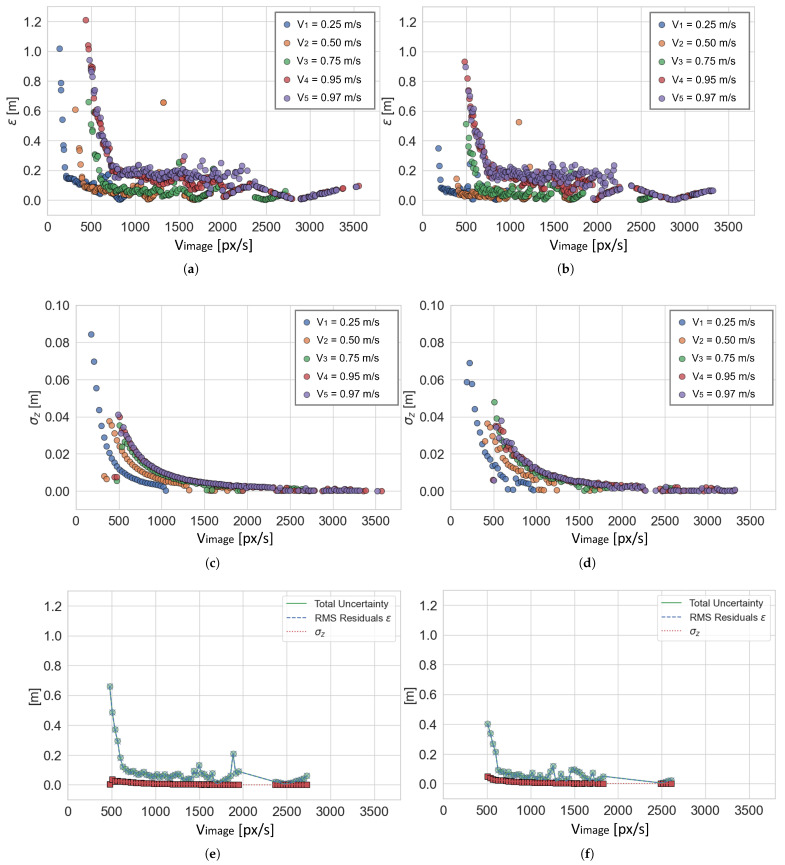
(**a**,**b**) Graphs showing the $RMS_{g,k}$ values for models with k=1…5 (no filtering) and with k=6…10 (filtering applied), respectively. (**c**,**d**) Graphs showing the $\sigma_{{\tilde{Z}}_{g,k}}$ values for models with k=1…5 (no filtering) and with k=6…10 (filtering applied), respectively. (**e**,**f**) Graphs showing, for $S_{3}=0.75$ m/s, the contribution of $RMS_{g,3}$ (blue dashed line with “X” markers) and $\sigma_{{\tilde{Z}}_{g,3}}$ (red dotted line with “squared” markers) towards the computation of the total uncertainty $u_{g,3}$ (green solid line with “circular” markers), for k=3 (no filtering) and k=8 (filtering applied), respectively.

**Table 1 sensors-25-03790-t001:** Summary of resulting generalized uncertainty for all tested robot speeds $S_{1}\ldots S_{5}$. Data is shown for both the original data ($u_{1-5}$) and for the data resulting from the window-based filtering procedure ($u_{6-10}$).

Robot Speed [m/s]	$u_{1-5}$ [m]	$u_{6-10}$ [m]
0.25	0.15	0.07
0.50	0.08	0.04
0.75	0.09	0.07
0.95	0.20	0.19
0.97	0.22	0.21

## Data Availability

The data set is available upon request from the authors. The developed code used for this work is made publicly available on GitHub (see reference [33]).

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
