# Peer review of "Depth from 2D Images: Development and Metrological Evaluation of System Uncertainty Applied to Agricultural Scenarios"

_sensors, 2025, doi:10.3390/s25123790_

Round 1

Reviewer 1 Report

Comments and Suggestions for Authors

Dear Authors,

The article present in detail analysis, the experimental system development, validation and uncertainty analysis for monocular depth estimation based on optical flow for agricultural vehicles. 

The paper is interesting to read and it covers the theoretical aspect of the modeling of depth estimation very well. The experimental setup is well presented and the validation and results are sufficient. 

Following are some directions/comments/ quires for the authors to improve the manuscript and further work.

  1. I felt the presented work can be expanded beyond agricultural purposes. If the authors wish to highlight that this work focuses more on agricultural machine navigation,  you should justify the choice of design parameters in related to agricultural applications. Although speed of the vehicle was set based on practical settings, other object placements in the design and the capture environment such as lighting needs to be justified.
  2. In computer vision Lidar is a technique use heavily for depth estimation. Could you mention/jusity the reason for camera based depth estimation in this application setting? Improve the introduction to highlight the choice of monocular depth estimation based on optical flow for agricultural vehicles. 
  3. What is the generalization capacity of your system to different deployment environment? I.e how will the performance differ based on the environment of the deployment?  The authors do present about the camera and  velocity choices for the vehicle.  Expand this discussion on other environmental parameters which can affect the performance. You can simply have this as a discussion topic under the subsection limitation of the model

Author Response

[Comment] I felt the presented work can be expanded beyond agricultural purposes. If the authors wish to highlight that this work focuses more on agricultural machine navigation,  you should justify the choice of design parameters in related to agricultural applications. Although speed of the vehicle was set based on practical settings, other object placements in the design and the capture environment such as lighting needs to be justified.

[Response] We agree with the reviewer; although the developed model was designed with the agriculture scenario in mind, it can be applied to everything that moves and the results observed are relevant in general. We note that we clearly stated that our setup was designed for agriculture scenarios in several points of the article (lines 54-59, 66-69, 85-94, Sections 3.1 and 3.2), as such design choices were made accordingly as already stated in the manuscript. However, we added a conclusive note at lines 500-509 to address the topic of generalization of the system to other scenarios as well, also incorporating the potential limitations of the system and other design choices to keep in mind in those cases.

[Comment] In computer vision Lidar is a technique use heavily for depth estimation. Could you mention/jusity the reason for camera based depth estimation in this application setting? Improve the introduction to highlight the choice of monocular depth estimation based on optical flow for agricultural vehicles. 

[Response] LiDAR sensing is the main technology adopted in the field of outdoor measurements, especially for agriculture. However, several issues arise: (1) accurate sensors are typically costly, (2) LiDAR devices acquire dense point clouds and as such need dedicated acquisition hardware, (3) they require the users to move at lower speeds to ensure correct data acquisition, (4) the sunlight can interefere with the acquisition creating abherrations. Not every application requires this much accuracy or even density of points, so a lightweight choice is proposed in this work leveraging only 2D cameras and a state-of-the-art common algorithm, optimized for the application. We incorporated this discussion in the Introduction Section as requested (lines 22-29).

[Comment] What is the generalization capacity of your system to different deployment environment? I.e how will the performance differ based on the environment of the deployment?  The authors do present about the camera and  velocity choices for the vehicle.  Expand this discussion on other environmental parameters which can affect the performance. You can simply have this as a discussion topic under the subsection limitation of the model.

[Response] We thank the reviewer for this suggestion. Indeed, other environmental parameters may impact the results of the proposed model, such as illumination, vibration, adverse metereological conditions. We expanded Section 3 with this discussion at lines 500-509.

Reviewer 2 Report

Comments and Suggestions for Authors

The authors raise the problem of depth estimation using a monocular chamber for agricultural machinery. The introduction clearly states the limitations of existing methods (instability of metrics, high resources, lack of metric coordinates) and positions the proposed solution as a simple alternative. To solve this problem, a combination of optical flow analysis and calculation of measurement uncertainty is used in the format of a generalized and integrated approach. The optimal camera movement speeds for each approach have been identified, providing the least uncertainty. The main value of the presented work lies in the formation of an analytical model and its experimental form, and its practical confirmation. In addition, a generalized and integrated approach for calculating uncertainty is valuable, as well as recommendations obtained during experiments on the choice of fps and camera speed.

The authors described the laboratory bench in sufficient detail, assembled a dataset of normal size (60 000 records), which considered both different speeds and directions of camera movement. The great advantage of the article is that the experimental code is publicly available, this will allow readers to study the experiment more deeply and modernize/ repeat it for their subject area. In addition, the authors validated the experimental and analytical models (Figure 6), and the results are comparable.

Conclusions are formulated according to the results of the experiment. The authors provide practical recommendations based on their own experience and experimental data.

The authors provided 39 references, the works relate to a period of 5-10 years, perhaps it is worth updating here. Comments:

1. Line 342 mentions a speed of "500-800 m/s", I think it meant pixels per second.

2. In Figure 8, the lower graphs show some gaps in the region of 2000-2400 pixels per second. Please explain this aspect in the data, as I did not find an explanation in the text.

3. Perhaps it would be worthwhile to estimate the processing time or the delay time between the receipt of data and the receipt of the result, this is an important aspect for real-time systems installed on equipment.

Otherwise, the article is written with high quality.

Author Response

[Comment] The authors provided 39 references, the works relate to a period of 5-10 years, perhaps it is worth updating here.

[Response] We thank the reviewer for this comment. We added a few more works related to the topics addressed that are more recent. Please note that some references are old and cannot be removed since they are the fundamentals of some algorithmic aspects of our work (e.g. the original Optical Flow references in ref. 2 and 3, other state-of-the-art references of computer vision techniques as in ref. 5, ref 24, ref 32, ref 36). We added a few more references published not later than 2022.

[Comment] Line 342 mentions a speed of "500-800 m/s", I think it meant pixels per second.

[Response] We thank the reviewer for noting this typo. We corrected the issue.

[Comment] In Figure 8, the lower graphs show some gaps in the region of 2000-2400 pixels per second. Please explain this aspect in the data, as I did not find an explanation in the text.

[Response] These gaps are due to missing detections caused by the high velocity of the robot and poor quality of the targets (too blurred or too light/dark). We addressed this issue in the text at lines 472-479.

[Comment] Perhaps it would be worthwhile to estimate the processing time or the delay time between the receipt of data and the receipt of the result, this is an important aspect for real-time systems installed on equipment.

[Response] We thank the reviewer for this comment. We think that the reviewer is referring to a sort of “inference time”, e.g. the time intercurring between the acquisition of the frames and the output being produced by the pipeline. This time is obtained for a certain pair of images by taking into account: (1) the acquisition speed of the camera and the storage capacity of the hardware to avoid potential bottlenecks during acquisition, (2) the time required by the processing software to detect the points of interest in the pairs of images, in our case the ArUco markers, (3) the time required by the software to compute OF on the point of interest detected, (4) the time required by the model to finally produce a depth estimation given the estimated OF value. The first point can be addressed by using better hardware (a faster camera, a better equipped computer). Points (3) and (4) are very quick to be computed since it’s a simple formulation applied on pre-processed data instead of images. The true bottleneck of the application is point (2), which depends on the point of interest to be detected. We used the ArUco library, requiring around 100 ms per image to produce an estimation since it’s based on computer vision techniques instead of AI-based. By adopting AI-based models (e.g. object detectors, segmentation algorithms, skeletonization algorithms), this time may increase up to 500 ms per image according to the size of the image to process. All this considered, we estimated this time equal to 300 ms on average for our set-up. This discussion has been incorporated at lines 480-499.